# Three-dimensional heads-up surgery in ab-interno trabeculotomy: Image processing-assisted trabeculotomy

**Takafumi Suzuki**[1,2], **Takashi Fujishiro**[1]*, **Koichiro Sugimoto**[1], **Makoto Aihara**[1]

1 Department of Ophthalmology, University of Tokyo Graduate School of Medicine, Tokyo, Japan,
2 Department of Ophthalmology, Shinseikai Toyama Hospital, Toyama, Japan

* fujishiro.tky@gmail.com

**Data Availability Statement:** All relevant data are within the manuscript and its Supporting Information files.

**Funding:** no specific funding for this work.

## Abstract

### Purpose

We compared the visibility and surgeon posture between image-processing-assisted trabeculotomy (IP-LOT) using the NGENUITY® 3D visual system and conventional microsurgery (microscope-assisted trabeculotomy; MS-LOT).

### Methods

IP-LOT was performed for five pig eyes. The visibility of the trabecular mesh work was evaluated on images of the trabecular mesh work and the posterior surface of the cornea (Cor) obtained under three different conditions. Images were then analyzed using ImageJ® to measure differences in luminance between the trabecular mesh work and Cor. IP-LOT was also performed for eleven human eyes, and the data were analyzed using the same approach as that used for the pig eyes. The length from the surgeon's abdomen to the operative eye (working distance) during MS-LOT and IP-LOT was measured for 12 different surgeons and compared to evaluate surgeon posture.

### Results

Image processing significantly increased the difference in luminance between the trabecular mesh work and Cor in both pig and human eyes ($p < 0.05$). Moreover, the working distance in IP-LOT was significantly shorter than that in MS-LOT ($p < 0.05$).

### Conclusion

Our findings suggest that the NGENUITY® 3D visual system provides better trabecular mesh work visibility than a normal microscope in conventional surgical methods, and it allows surgeons to operate without moving far from the operative eye.

**Competing interests:** no competing interests exist.

## Introduction

Glaucoma is defined as a disease presenting with functional and structural abnormalities of eyes with characteristic changes in both the optic nerve papilla and the visual field, and the optic neuropathy can be corrected or controlled by lowering the intraocular pressure (IOP). The pathogenesis of glaucoma involves glaucomatous optic neuropathy (GON) with chronic progressive retinal ganglion cell death and corresponding visual field abnormalities [1].

IOP reduction is considered the only reliable, evidence-based treatment for glaucoma and has been reported to suppress the onset and progression of glaucoma, regardless of its type or stage [2–9]. IOP-reducing treatment modalities for glaucoma include administration of eye drops or oral medications; laser therapy, such as SLT and CPC; and surgery, including trabeculectomy, trabeculotomy, or tube shunt implantation. Minimally invasive glaucoma surgery (MIGS), which is less invasive and safer than conventional surgery, has become a common procedure for outflow surgery [8–30].

The typical MIGS involves ab interno insertion of implant devices or trabecular incision with a hook or abrasion. The procedure for outflow MIGS begins by tilting the patient's face and eye in the direction of the trabecular meshwork (TM) to be incised, and tilting the microscope in the opposite direction. A surgical gonioprism, as well as viscoelastic substances, are placed on the cornea to observe the TM.

In the field of MIGS, the surgeon needs to be in a tilted position away from the operating eye while looking through the microscope; thus, this procedure is difficult for the surgeon. Recently, head-up surgery (HUS) using a 3D microscope has been developed and widely used in the fields of vitreous surgery (e.g., ERM, MH, and RRD) and cataract surgery [31–40]. The retinal toxicity caused by transillumination is more problematic in conventional vitrectomy and cataract surgery than in glaucoma surgery [31, 35, 36]. In addition, in HUS using a 3D microscope, assistants and other people around the surgeon, as well as the surgeon, can share the same field of view on a large screen with high resolution. This makes it easier to provide surgical guidance and education. In addition, the ability for image processing, e.g., changing the brightness and contrast, allows surgery to be performed under low transillumination, thereby reducing retinal toxicity [31, 35, 36]. Other advantages include a wider depth of focus even with strong magnification, reduction of neck and back pain by improving the postural restrictions for the surgeon, and increased accuracy of surgical operation [31–40] (Table 1). Based on these considerations, we speculated that ab-interno trabeculotomy performed with HUS using 3D microscopy is also highly useful. In this study, we performed image processing-assisted trabeculotomy (IP-LOT) with HUS using the NGENUITI® 3D visualization system and compared the visibility and surgeon's posture between IP-LOT and the conventional microscopic method (microscope-assisted trabeculotomy: MS-LOT).

## Methods

### Equipment for visualization

We performed IP-LOT using the NGENUITY 3D® Visualization System with the VERION Image Guided System (Alcon, Fort Worth, TX, USA), which includes a 3-D high dynamic range (HDR) camera with a complementary metal-oxide-semiconductor image sensor, specialized software (TrueWare, version 9.5.4; TrueVision Systems, Inc., Santa Barbara, CA, USA), and a high-definition 55" LCD monitor with a 4K OLED display (LG, Seoul, Republic of Korea) that uses passive 3-D display technology (Fig 1).

For the best 3D effect, the display was positioned next to the operative eye and 4–6 feet (1.2. 1.8 m) away from the surgeon. The screen was at the same height as the surgeon when the

**Table 1. Summary of advantages and disadvantages of vitrectomy and cataract surgery using 3D microscopy.**

| | Head up surgery | Normal microscope surgery |
|---|---|---|
| **Surgeon posture** | Comfortable position | The surgeon may experience neck and spine pain. |
| **Surgical assistant** | Operates while looking at the monitor | Operates while looking through a microscope |
| **Visibility** | Improved by magnification and image processing | Improved with only magnification. |
| **Depth of focus** | Maintained even with magnified vision. | Shallower with magnified vision. |
| **Retinal toxicity** | Reduced endoillumination with image processing. | No reduction. |
| **Cost** | Special microscope and 3D monitor. | Normal microscope without a monitor. |
| **Required space** | Space required to place a 3D monitor | No space required for a monitor. |
| **Surgical view** | Everyone can share the same vision. | Different views for the surgeon and non-surgeons. |
| **Technical aspect** | Takes some time to get used to. | No need to get used to. |
| **Other aspects** | Improved visibility of ILM and reduced dye. Visible even in cases of intermediate translucent body opacity such as vitreous hemorrhage | |

surgeon was seated at the operating table. We adjusted the height, tilt, and distance of the display at the discretion of the doctor to provide the best image.

## Surgical procedure

The microscope used for the surgery was the NGENUITY 3D® Visualization System with VERION Image Guided System for IP-LOT and LUXOR®LX3 for MS-LOT. IP-LOT and MS-LOT were performed in pig eyes fixed on a Styrofoam face model for surgical practice. IP-LOT was performed on human eyes.

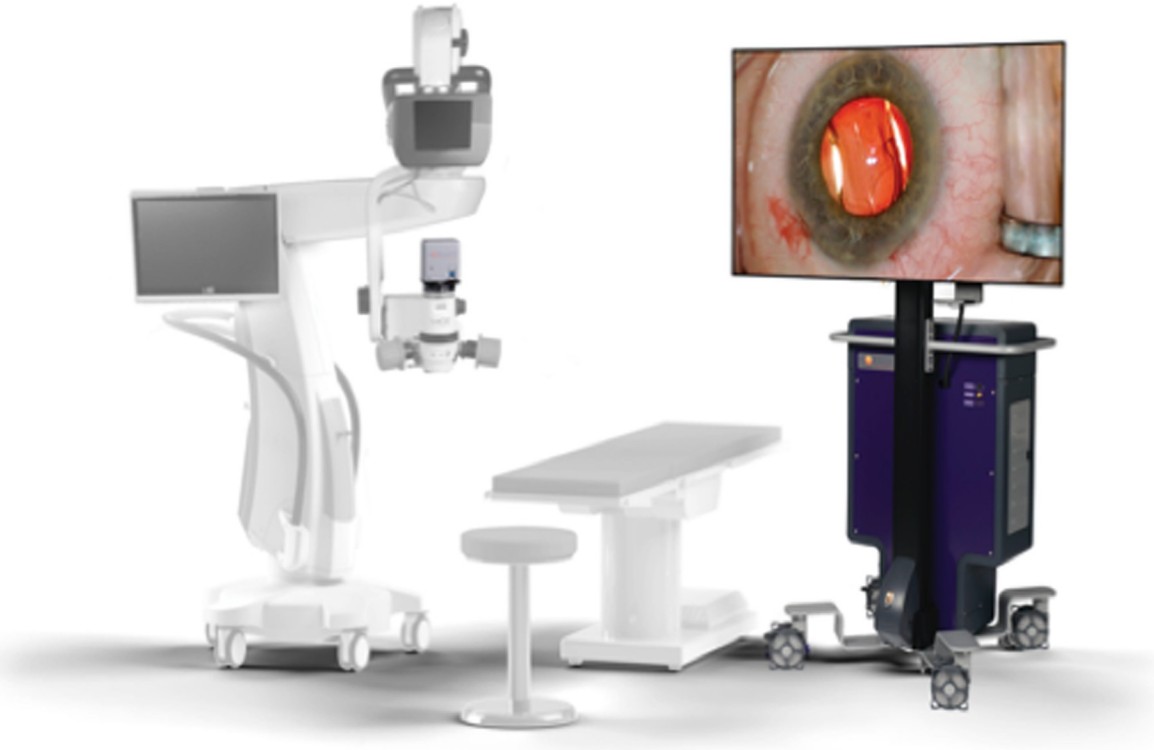

**Fig 1. Pictures of the microscope and display monitor.** The operating microscope (left) with the image capture module (ICM) sends up to 3 GB image information per second to the embedded processing unit (EPU), and the sent image was projected onto the 3D display (right).

In the IP-LOT, the surgeon wore special 3D glasses and created a main wound in the 12 o'clock direction of the operative eye with a 2.4-mm slit knife while looking only at the monitor. A sufficient amount of viscoelastic substance was injected into the anterior chamber to allow observation of the angle within 180˚ contralateral to the main wound.

The styrofoam model of the face for pig eye surgery and the real face for human eye surgery were tilted in the direction of the angle to be observed, and the microscope was tilted in the direction that agitated the angle as much as possible. The Tanito ab-interno Trabeculotomy Micro-hook (Inami, Tokyo, Japan) was inserted through the main wound while watching the monitor. The base of a surgical gonioprism (Swan-Jacob Autoclavable Gonioprism; Ocular Instruments Inc., Washington, USA) was then placed on the cornea with its base turned to an angle that was opposite to the main wound, and the TM was observed on a monitor.

In the MS-LOT, the main wound was created under the microscope, and viscoelastic substances were fully injected into the anterior chamber. The facial model and microscope were tilted in the same way. Then, the Tanito ab-interno Trabeculotomy Micro-hook was inserted through the main wound again under the microscope, and the base of the surgical gonioprism (Swan-Jacob Autoclave Gonioprism) was placed on the cornea with its base opposite to the main wound. The TM was then observed.

As for the pig's eye, we used pig's eyes for food obtained from Shibaura Slaughterhouse Tokyo Metropolitan Government (Tokyo, Japan). In addition, all animal experiments were performed according to the ethical guidelines for animal experimentation of the University of Tokyo (approval number: 1795) and the ARVO Statement for the Use of Animals in Ophthalmic and Vision Research.

The observational study on humans was approved by the Ethics Committee of the University of Tokyo Hospital (approval number: 2270) and was conducted in accordance with the principles of the Declaration of Helsinki. Informed consent (the medical treatment itself and the use of their medical data for research) was obtained in writing from all human patients prior to the surgeries, and that for our pig surgeries was obtained from all surgeons in the same way.

## Visibility evaluation: Measurement of luminance at the angle under various conditions

First, 5 pig eyes were subjected to IP-LOT. For each eye, IP-LOT was performed at three different locations within 180˚ opposite to the main wound. Each location was 60˚ away from the others. A total of 15 photographs of the angles were taken.

To evaluate the visibility of the TM during surgery, the following three conditions were used for obtaining images. No image processing indicates the absence of both contrast and exposure adjustments (brightness, 47.8 cd/m$^2$; contrast, 54.9 arb. units). Condition 1 involved mild contrast enhancement and mild exposure reduction (brightness, 40.5 cd/m$^2$; contrast, 65.2 arb. units), and condition 2 involved intense contrast enhancement and intense exposure reduction (brightness, 40.3 cd/m$^2$; contrast 73.8 arb. units). We defined the visibility in MS-LOT as that without image processing and in IP-LOT as with image processing.

Using ImageJ®, we calculated the difference in luminance between the TM and Cor (0.5 mm long x 3.5 mm wide) and compared the mean values of the differences in luminance for each eye (Fig 2). The luminance was measured on a 256-step scale from 0 to 255, where 0 represents black and 255 represents white. The measurements are presented in an arbitrary unit.

Next, 11 human eyes were included in the study. For each human eye, IP-LOT was applied to a single region of the TM from the inferior to the nasal inferior area, and imaging was performed under the three conditions. No image processing' indicates no contrast or exposure

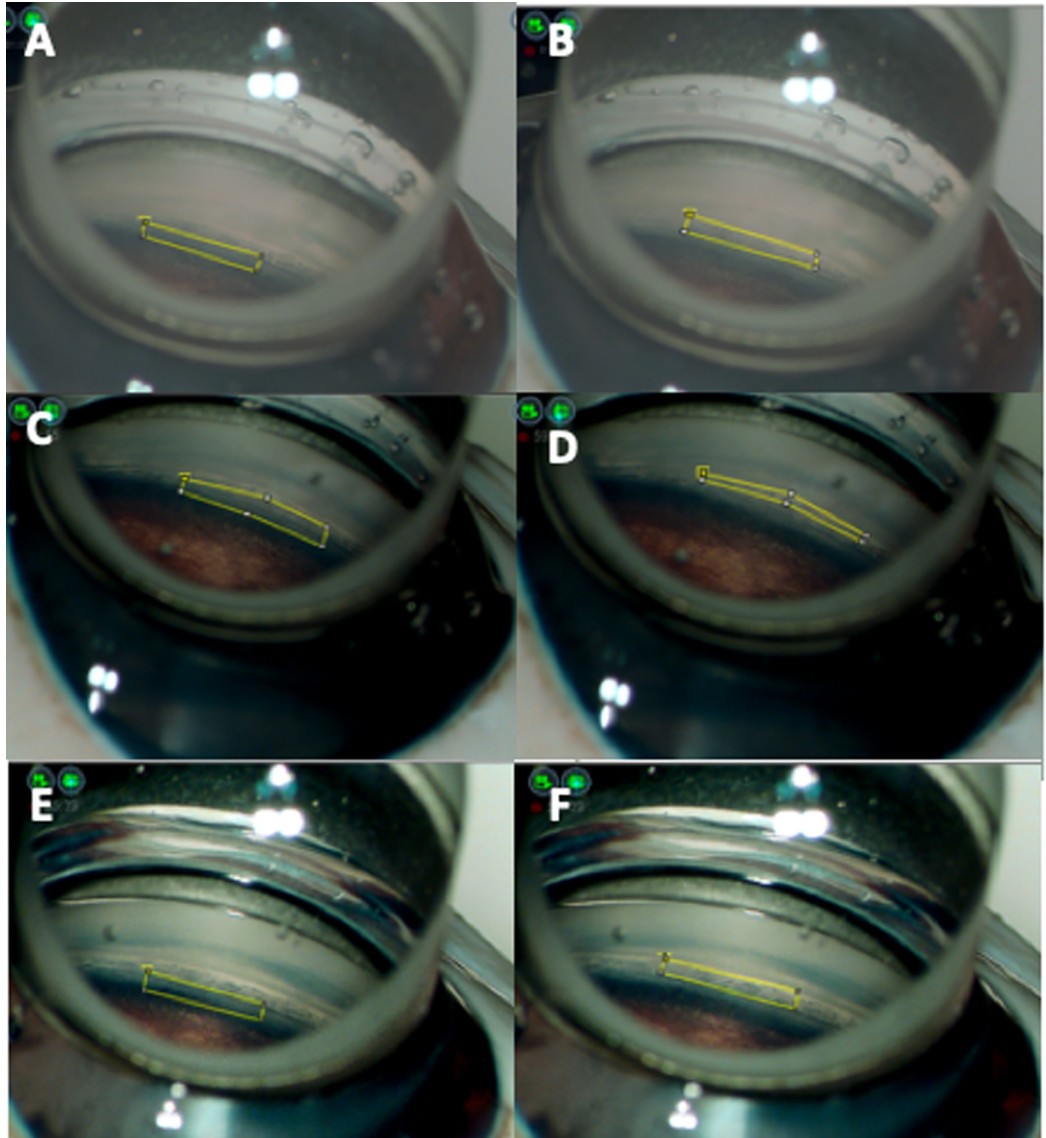

**Fig 2. Images of the angle in pig eyes under various conditions and examples of measurements of luminance with ImageJ®.** (A) Trabecular meshwork (TM) area (without image processing). (B) Posterior corneal (Cor) area (without image processing). (C) TM area (Condition 1). (D) Cor area (Condition 1). (E) TM area (Condition 2). (F) Cor area (Condition 2). The imaging conditions for NGENUITY® are described as follows. Both brightness and contrast were changed for each condition. Without image processing: The brightness was 47.8 cd/m$^2$, and the contrast was 54.9 arb. units. Condition 1: The brightness was reduced from 47.8 cd/m$^2$ to 40.5 cd/m$^2$, and the contrast was increased from 54.9 arb. units to 65.2 arb. units. Condition 2: The brightness was reduced from 47.8 cd/m$^2$ to 40.3 cd/m$^2$, and the contrast was increased from 54.9 arb. units to 73.8 arb. units. The 0.5 mm × 3.5 mm area of the angle was used as the analysis area in all images.

adjustment (brightness, 47.8 cd/m$^2$; contrast, 54.9 arb. unit). Condition 1' involved mild contrast enhancement (brightness, 47.8 cd/m$^2$; contrast, 65.5 arb. unit), and condition 2' involved intense contrast enhancement (brightness, 47.8 cd/m$^2$; contrast, 75.5 arb. unit). As performed in pig eyes, we used ImageJ® to determine the difference in luminance between TM and Cor in the images under the three conditions and compared the mean of differences in luminance under these conditions (Fig 3).

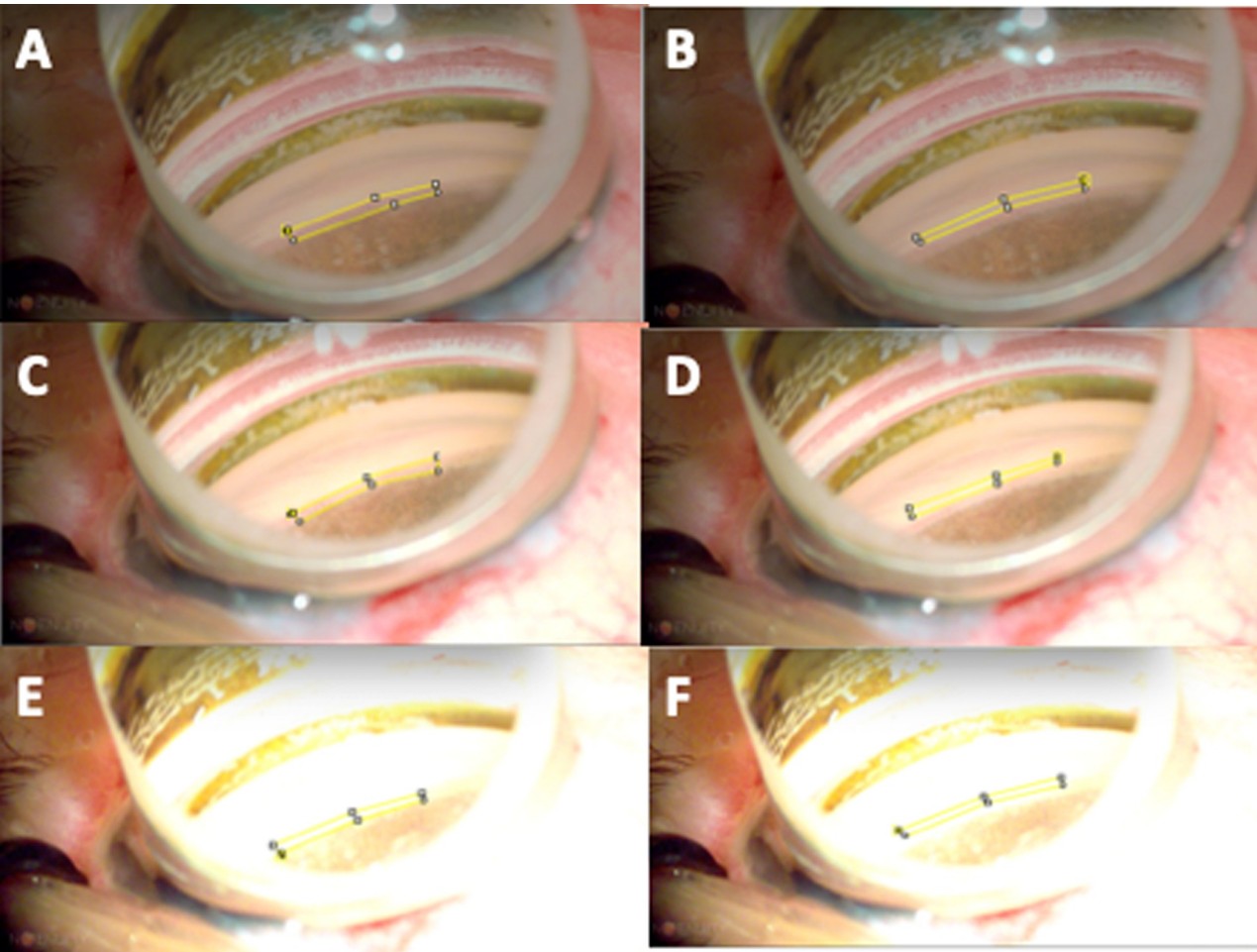

**Fig 3. Images of the angle in human eyes under various conditions and examples of measurements of luminance with ImageJ®.** (A) Trabecular meshwork (TM) area (without image processing). (B) Posterior corneal (Cor) area (without image processing). (C) TM area (Condition 1'). (D) Cor area (Condition 1'). (E) TM area (Condition 2'). (F) Cor area (Condition 2'). The imaging conditions of NGENUITY® are described as follows. Only the contrast was changed for each condition. Without image processing': The brightness was 47.8 cd/m$^2$, and the contrast was 54.9 arb. units. Condition 1': Brightness was kept at 47.8 cd/m$^2$, and contrast was increased from 54.9 arb. units to 65.5 arb. units. Condition 2': Brightness was kept at 47.8 cd/m$^2$, and contrast was increased from 54.9 arb. units to 75.5 arb. units. The 0.5 mm × 3.5 mm area of the angle was used as the analysis area in all images.

### Assessment of the surgeon's posture: Measurement of working distance

In order to evaluate the posture of the surgeon, we measured and compared the distance from the surgeon's abdomen to the operative eye (working distance) in the MS-LOT and IP-LOT for 12 different surgeons when performing LOT for a pig eye (Fig 4).

### Statistical analysis

Differences in luminance between the TM and Cor under each imaging condition in the pig and human eyes were evaluated. In the pig eyes, the mean value and standard deviation for each condition were calculated and compared using the paired t-test, with a significance level of $p < 0.05$. In addition, the mean value and standard deviation of working distance for MS-LOT and IP-LOT were calculated, and the differences by surgical technique were evaluated using the paired t-test. The significance level was set at $p < 0.05$. In contrast, the median

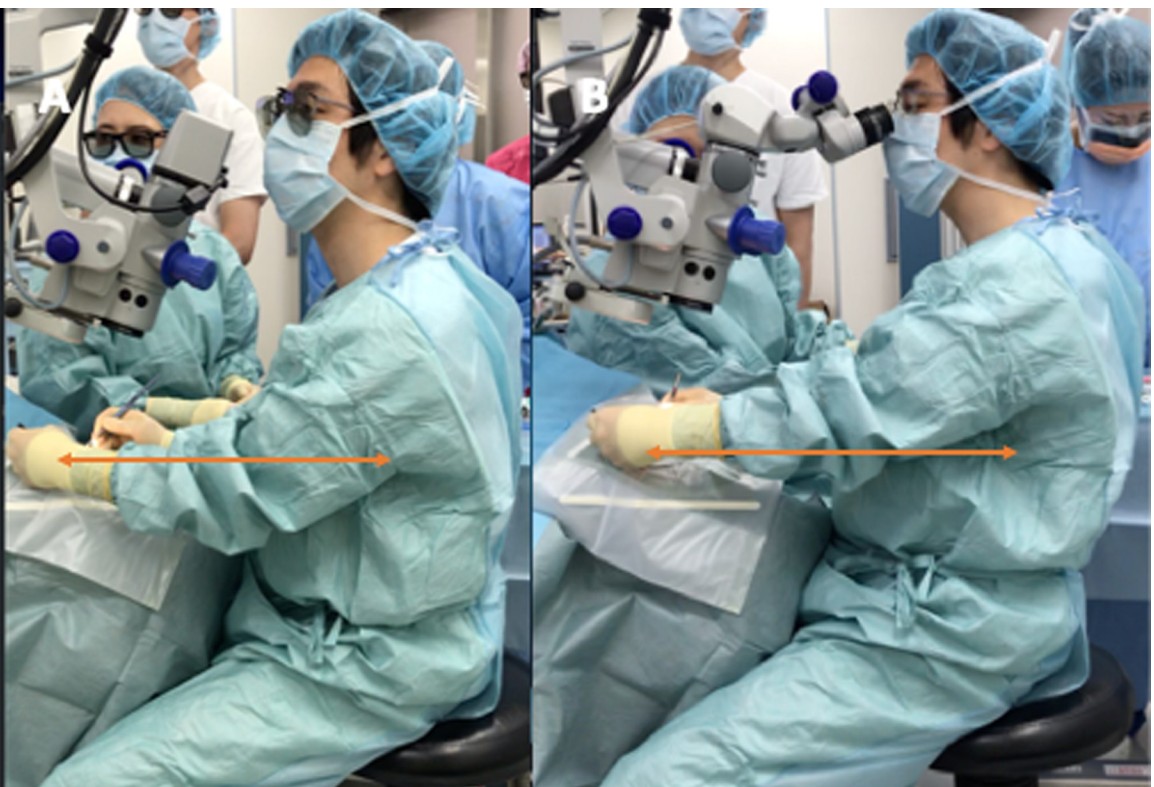

**Fig 4. Example of surgeon posture and measurement of working distance in microscope-assisted trabeculotomy (MS-LOT) and image-processing-assisted trabeculotomy (IP-LOT).** (A) Surgeon's posture in MS-LOT. (B) Surgeon's posture in IP-LOT. To evaluate the working distance, the distance from the operative eye to the surgeon's abdomen was measured, as indicated by the length of the arrow above.

value and interquartile range for each imaging condition were calculated and compared using the Wilcoxon signed-rank test, with a significance level of $p < 0.05$ because of the variation of the measured values in the human eyes.

## Results

### Visibility evaluation: Measurement of luminance at the angle under various conditions

First, we summarized the luminance values for the TM and the posterior surface of the cornea (Cor) in five pig eyes without processing, under condition 1, and under condition 2 (Table 2).

The differences in luminance (means and standard deviations) were 30.9 ± 2.6 arbitrary units (arb. units) without processing, 57.5 ± 1.7 arb. units under condition 1, and 86.0 ± 9.4 arb. units under condition 2. Image processing increased the difference in luminance between TM and Cor, and significant differences were observed between no processing and condition, and between no processing and condition 2 ($p < 0.05$) (Fig 5).

Next, we summarized the TM and Cor luminance values in eleven human eyes without processing', under condition 1', and under condition 2' (Table 2). The median (interquartile range) differences in luminance were 18.8 (15.8–21.2) arb. units without processing, 21.8 (21.0–27.4) arb. units under condition 1', and 8.5 (4.3–14.7) arb. units under condition 2'. Under condition 1', image processing significantly increased the differences in luminance between TM and Cor ($p < 0.05$). However, the difference in luminance did not significantly

**Table 2. Differences in luminance in angles in 5 pig eyes and 11 human eyes under each condition.**

|  | N | C 1 | C 2 |  | N' | C1' | C2' |
|---|---|---|---|---|---|---|---|
| **Mean (arb. units)** | 30.9 | 57.5 | 86.0 | **Median (arb. units)** | 18.8 | 21.8 | 8.5 |
| **Standard deviation** | 2.6 | 1.7 | 9.4 | **Interquartile range** | 15.8–21.2 | 21.0–27.4 | 4.3–14.7 |

N, No processing: Brightness, 47.8 cd/m$^2$; Contrast, 54.9 arb. units.

C1, Condition 1: Brightness, 40.5 cd/m$^2$; Contrast, 65.2 arb. units.

C2; Condition 2: Brightness, 40.3 cd/m$^2$; Contrast, 73.8 arb. units.

N', No processing': Brightness, 47.8 cd/m$^2$; Contrast, 54.9 arb. units.

C1', Condition 1': Brightness, 47.8 cd/m$^2$; Contrast, 65.5 arb. units.

C2', Condition 2': Brightness, 47.8 cd/m$^2$; Contrast, 75.5 arb. units.

increase under Condition 2' compared to without processing' (Fig 5). All surgeries were completed without major complications.

## Evaluation of the surgeon's posture: Measurement of working distance

The working distances (mean values and standard deviations) significantly differed between MS-LOT (42.4 ± 2.8 cm) and IP-LOT (33.8 ± 2.4 cm) ($p < 0.05$) (Table 3) and were significantly shorter in the IP-LOT than in the MS-LOT (Fig 5).

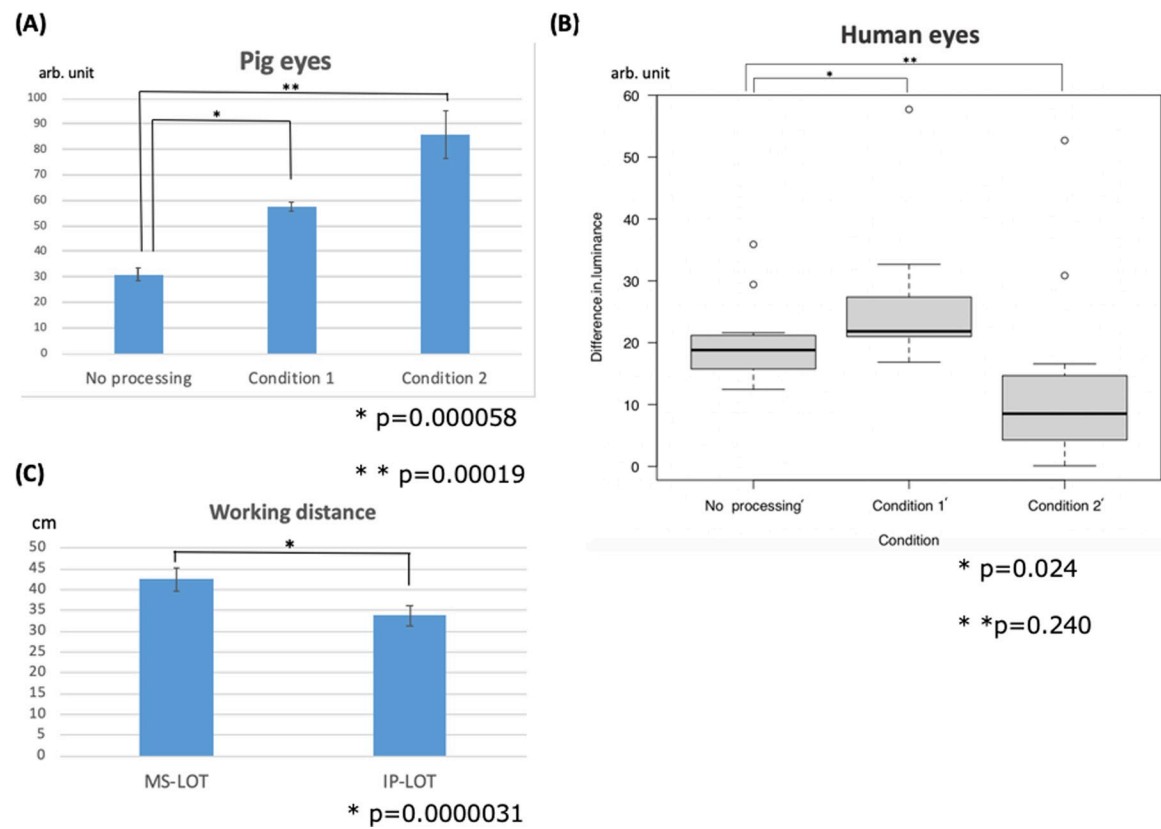

**Fig 5. The average difference in luminance in pig and human eyes under each condition and in the working distances.** (A) Image-processing-assisted trabeculotomy (IP-LOT) for 5 pig eyes. There were significant differences between no processing and condition 1 and between no processing and condition 2 (n = 5, $p < 0.05$). (B) IP-LOT for 11 human eyes. There were significant differences between no processing' and condition 1' (n = 11, $p < 0.05$). However, there were no significant differences between no processing' and condition 2'. (C) Working distance in microscope-assisted trabeculotomy (MS-LOT) and IP-LOT. The working distance in IP-LOT was significantly shorter than that in MS-LOT (n = 12, $p < 0.05$).

**Table 3. Working distance in 12 surgeons.**

|  | Microscope-assisted trabeculotomy | Image-processing-assisted trabeculotomy |
|---|---|---|
| **Mean (cm)** | 42.4 | 33.8 |
| **Standard deviation** | 2.8 | 2.4 |

## Discussion

In comparison with traditional surgery, the current MIGS is safer with fewer complications and is expected to be equally or more effective [24, 26]. MIGS is indicated not only for patients with inadequate IOP reduction but also for those with poor adherence [24, 26]. In addition, surgical devices, including the Kahook Dual Blade or Tanito ab-interno Trabeculotomy Micro-hook [8, 11–19] and the iStent trabecular micro-bypass [24, 26, 28], have become popular in recent years. However, during surgery, the visibility of the TM differs from case to case, and identification of the TM is sometimes difficult. In addition, the surgeon has to tilt the microscope and move away from the operating eye during surgery, consequently forcing the surgeon into an unnatural posture. While the use of Mori goniotomy lenses (R E MEDICAL Inc., Osaka, JAPAN) allows visualization of the angle directly above and may thus solve the problem of posture [41], the issue of visibility remains unresolved.

To date, only two previous reports have described iStent MIGS performed by HUS using 3D microscopy [42, 43], but neither study included cases involving ab-interno trabeculotomy. The first study assessed 74 eyes in 56 patients; 72 eyes underwent cataract surgery, and 2 eyes were treated with vitrectomy alone. Among the 72 eyes with cataract surgery, 60 eyes underwent cataract surgery alone, 7 eyes underwent cataract surgery and iStent insertion, and 5 eyes underwent cataract surgery and vitrectomy. All surgeries were completed without major complications. Although the type of gonioscope used during surgery was not described, the authors reported that the surgeon did not have to change posture during the surgery and could safely perform the surgery with low illumination and brightened images [42]. That report mainly described how HUS reduced the transillumination-induced retinal toxicity in cataract surgery and vitrectomy. However, that study did not perform quantitative and comparative evaluations or statistical analyses of the surgeon's posture. In addition, the study included only 7 cases of glaucoma surgery with the iStent in combination with cataract surgery, and it did not mention the visibility of the corneal angle during insertion of the iStent. In the second report, the surgeon performed cataract surgery with the Toric IOL and iStent with HUS using Ocular Hill Surgical Gonioprism (Ocular Instruments Inc., Washington, USA). That study found that the iStent could be placed in the same posture as that used during the cataract surgery without the need to change the surgeon's posture during the surgery, and the increased depth of focus of the 3D microscope, even at high magnifications, eliminated the need for frequent focusing. In addition, the study also showed that retinal toxicity could be reduced by performing surgery with low transillumination [43]. However, that study did not clarify the sample size. Moreover, although intraoperative image processing was performed to improve visibility, the author did not clarify the changes made to the imaging conditions or perform any quantitative and comparative evaluations or statistical analysis of visibility. The study also showed that the surgery could be performed without changing the surgeon's posture during the operation; however, that study did not perform quantitative and comparative evaluations or statistical analysis of the surgeon's posture either. Because these two studies did not evaluate the optical improvement in visibility due to the use of HUS in iStent and combined cataract surgery, we evaluated the difference in luminance between the TM and Cor as an indicator of the optical angle visibility provided by HUS because the imaging processor can enhance the contrast of the

obtained image through the CCD camera instead of the direct vision through the microscope. In addition, since these two studies did not evaluate the surgeon's posture quantitatively, we also evaluated the difference in the working distance as a factor of the surgeon's posture.

Visibility was defined as the difference in luminance between the TM and Cor, and luminance was compared using ImageJ$^{®}$. Image processing significantly increased the difference in luminance for both pig and human eyes. In other words, we showed that TM visibility could be improved by changing the imaging conditions. However, as shown in Fig 3 (condition 2'), when the contrast was increased too much, the entire image was blown out to white; accordingly, the visibility of the image did not improve at one glance. In fact, the differences in luminance under condition 2' did not significantly increased compared with that without image processing (Fig 5). This finding suggests the importance of selecting appropriate conditions and indicates that the visibility evaluation method considering the difference in luminance between the TM and Cor correlates well with the visibility actually experienced by the surgeon.

As for the working distance, the previous study on HUS [42] only showed that the surgeon could perform the surgery without changing the posture. In this study, we examined changes in the surgeon's posture by measuring changes in the distance between the operating eye and the surgeon's abdomen. The reason for the difference in working distance between the MS-LOT and IP-LOT is that the microscope with the lens barrel was tilted toward the surgeon and the surgeon had to look into it. Our findings suggest that the 3D microscope is useful for performing ab-interno trabeculotomy while the surgeon maintains a comfortable distance between the surgeon and the operated eye. Another advantage of HUS is that the surgical field of view can be shared. This is highly useful from an educational standpoint because the supervising physician can provide real-time surgical guidance with the screen.

This study has several limitations. First, the pig eyes did not reproduce the bleeding and clouding of the gonioscope due to body temperature, which are problems often encountered during real human surgery. Second, in order to find the optimal imaging condition, it is necessary to use the same imaging condition for both human eyes and pig eyes. However, for safety in the human surgery, the surgeon had to perform the TM incision under the conditions that he felt were easiest for looking at the TM (only contrast-enhanced condition 1', which was different from condition 1 used in pig eyes). In addition, the evaluations of the distance between the surgeon and the eye and the surgeon's posture were solely based on the distance from the surgeon's abdomen to the eye. Thus, this evaluation method might not be accurate for postural changes caused by surgical procedures.

As for future prospects, we expect that further developments in the image-processing technology used by microscopes will improve the possibility of performing surgeries safely by changing the image settings, which will improve visibility even in difficult situations, such as corneal opacity or anterior chamber hemorrhage during surgery.

## Supporting information

**S1 File.**
(XLSX)

## Acknowledgments

We would like to thank Editage (www.editage.com) for English language editing.

## Author Contributions

**Data curation:** Takafumi Suzuki, Takashi Fujishiro, Koichiro Sugimoto, Makoto Aihara.

Formal analysis: Takafumi Suzuki.

Investigation: Takafumi Suzuki, Takashi Fujishiro, Makoto Aihara.

Software: Takafumi Suzuki.

Writing – original draft: Takafumi Suzuki.

Writing – review & editing: Takashi Fujishiro, Koichiro Sugimoto, Makoto Aihara.

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
