## [Decision Letter · Decision Letter 0]

19 Oct 2021

PONE-D-21-18797Three-dimensional heads-up surgery in ab-interno trabeculotomy: Image processing-assisted trabeculotomyPLOS ONE

Dear Dr. Fujishiro,

Thank you for submitting your manuscript to PLOS ONE. After careful consideration, we feel that it has merit but does not fully meet PLOS ONE’s publication criteria as it currently stands. Therefore, we invite you to submit a revised version of the manuscript that addresses the points raised during the review process.

We look forward to receiving your revised manuscript.

Kind regards,

Aparna Rao

Academic Editor

PLOS ONE

Journal Requirements:

2. Please clarify the nature of the written informed consent in your Ethics Statement and Methods section. Did patients consent to the medical treatment, and/or did they specifically consent to participate in this study, including the use of their medical data for research.

Reviewers' comments:

Reviewer's Responses to Questions

**Comments to the Author**

1. Is the manuscript technically sound, and do the data support the conclusions?

Reviewer #1: No

2. Has the statistical analysis been performed appropriately and rigorously? 

Reviewer #1: No

3. Have the authors made all data underlying the findings in their manuscript fully available?

Reviewer #1: No

4. Is the manuscript presented in an intelligible fashion and written in standard English?

Reviewer #1: No

5. Review Comments to the Author

Reviewer #1: Thanks for submitting the manuscript

However there few comments for this

1. In page 12, the experimental conditions explained for both pig and human eyes are different. Could you please explain the difference in testing conditions if the new model of surgery has to be implemented then testing conditions need to be same for all practical purposes.

2. Table 2 results – in Human eyes the mean and standard deviation provided in table appears to be statistically not viable as standard deviation is more than or equal to 1/3 rd of mean. Eg The differences in luminance (mean values and standard deviations) were 21.1 ± 7.3 arb. units without processing', 28.1 ± 12.1 arb. units under condition 1', and 17.5 ± 15.6 arb. units under condition 2'.

3. Has the data been checked for normality before applying paired t test –seems inappropriate as the standard deviation is > 1/3 of mean . A non-parametric test would have been appropriate

4.Swan Jacob lens spelling incorrectly spelt in page 9 line 123

5. corner angle should have been corneal angle in page 19 line 282

6. PLOS authors have the option to publish the peer review history of their article (what does this mean?). If published, this will include your full peer review and any attached files.

Reviewer #1: No

---

## [Author Response · Author response to Decision Letter 0]

19 Nov 2021

Journal Requirements:

and https://journals.plos.org/plosone/s/file?id=ba62/PLOSOne_formatting_sample_title_authors_affiliations.pdf.

Response: 

Thank you for the comment. We have revised the the manuscript and file naming to meet PLOS ONE’s style. 

2. Please clarify the nature of the written informed consent in your Ethics Statement and Methods section. Did patients consent to the medical treatment, and/or did they specifically consent to participate in this study, including the use of their medical data for research.

Response: 

Thank you for the comment. The patients agreed to receive the medical treatment and to have treatment data used for research. We have added the following in the Methods section: “(the medical treatment itself and the use of their medical data for research)” (page 10, line 131–132).

Response: 

Thank you for the comment. Our datasets do not have any restrictions. Thus, we have uploaded the minimal anonymized dataset such as the measured values of the luminance in pig eyes and human eyes, and the working distance in Excel format as Supporting Information files.

 

Reviewer #1: 

1. In page 12, the experimental conditions explained for both pig and human eyes are different. Could you please explain the difference in testing conditions if the new model of surgery has to be implemented then testing conditions need to be same for all practical purposes.

Response: 

Thank you for the comment. We apologize for misleading you about the main outcome.

The main outcome of the current study was not to find a universal optimal image condition, but to show whether image processing can improve visibility.

As you pointed out, in order to find the optimal image condition, it is necessary to use the same imaging condition for both human eyes and pig eyes. However, for safety in the human surgery, the surgeon had to perform the TM incision under the conditions that he felt were easiest for looking at the TM (only contrast-enhanced condition 1’, which was different from condition 1 used in pig eyes). For comparison, we further increased the contrast and took images in condition 2’. We then examined whether the difference in luminance, which is an indicator of visibility, improved under these three types of conditions.

We have deleted the sentence “In the future, we hope to develop optimal microscope image settings for various situations” at the end of the Discussion section.

We added the following limitation in the Discussion section: “Second, in order to find the optimal imaging condition, it is necessary to use the same imaging condition for both human eyes and pig eyes. However, for safety in the human surgery, the surgeon had to perform the TM incision under the conditions that he felt were easiest for looking at the TM (only contrast-enhanced condition 1’, which was different from condition 1 used in pig eyes)” (page 22, line 334–338).

2. Table 2 results – in Human eyes the mean and standard deviation provided in table appears to be statistically not viable as standard deviation is more than or equal to 1/3 rd of mean. Eg The differences in luminance (mean values and standard deviations) were 21.1 ± 7.3 arb. units without processing', 28.1 ± 12.1 arb. units under condition 1', and 17.5 ± 15.6 arb. units under condition 2'.

Response: 

Thank you for the comment. As you pointed out, we think the parametric test was not suitable because of the high variability of the data in human eyes. Therefore, we increased the number of cases from 8 cases to 11 cases to reduce the variability and then performed a non-parametric test. Just like before, the results showed that there was a significant difference between No processing’ and Condition 1', and no significant difference between No processing’ and Condition 2'. We have also changed the notation to be median and interquartile range instead of mean ± standard deviation. To reflect the previous description, we have changed the number of cases from 8 to 11 in the corresponding parts of the main manuscript. In addition, we have changed the contents of Table 2 (page 15) and Figure 5, and added a sentence in the Methods section as follows: “In contrast, the median value and interquartile range for each imaging condition were calculated and compared using Wilcoxon signed-rank test, with a significance level of p < 0.05 because of the variation of the measured values in the human eyes” (page 15, lines 209–212). We have also changed the values in the Results section as follows: “The median (interquartile range) differences in luminance were 18.8 (15.8–21.2) arb. units without processing, 21.8 (21.0–27.4) arb. units under condition 1', and 8.5 (4.3–14.7) arb. units under condition 2'” (page 16, lines 235–239).

3. Has the data been checked for normality before applying paired t test –seems inappropriate as the standard deviation is > 1/3 of mean. A non-parametric test would have been appropriate.

Response: 

Thank you for the suggestion. As you pointed out, in human eyes, we agree that nonparametric tests are suitable because of the variation of the measured values. Therefore, the Wilcoxon signed-rank test was performed after we increased the number of cases. We added the following information in the Methods section: “In contrast, the median value and interquartile range for each imaging condition were calculated and compared using the Wilcoxon signed-rank test, with a significance level of p < 0.05 because of the variation of the measured values in the human eyes” (page 15, lines 209–212).

4. Swan Jacob lens spelling incorrectly spelt in page 9 line 123.

Thank you for the comment. We have revised the spelling.

5. corner angle should have been corneal angle in page 19 line 282.

Thank you for the comment. We have revised the spelling.

---

## [Decision Letter · Decision Letter 1]

24 Jan 2022

Three-dimensional heads-up surgery in ab-interno trabeculotomy: Image processing-assisted trabeculotomy

PONE-D-21-18797R1

Dear Dr. Fujishiro,

We’re pleased to inform you that your manuscript has been judged scientifically suitable for publication and will be formally accepted for publication once it meets all outstanding technical requirements.

Kind regards,

Aparna Rao

Academic Editor

PLOS ONE

Additional Editor Comments (optional):

Reviewers' comments:

Reviewer's Responses to Questions

**Comments to the Author**

1. If the authors have adequately addressed your comments raised in a previous round of review and you feel that this manuscript is now acceptable for publication, you may indicate that here to bypass the “Comments to the Author” section, enter your conflict of interest statement in the “Confidential to Editor” section, and submit your "Accept" recommendation.

Reviewer #1: All comments have been addressed

2. Is the manuscript technically sound, and do the data support the conclusions?

Reviewer #1: Yes

3. Has the statistical analysis been performed appropriately and rigorously? 

Reviewer #1: Yes

4. Have the authors made all data underlying the findings in their manuscript fully available?

Reviewer #1: Yes

5. Is the manuscript presented in an intelligible fashion and written in standard English?

Reviewer #1: Yes

6. Review Comments to the Author

Reviewer #1: thanks for addressing to the comments.The manuscript can be further improvised by removing redundant words. please look into it in future

7. PLOS authors have the option to publish the peer review history of their article (what does this mean?). If published, this will include your full peer review and any attached files.

Reviewer #1: No

---

## [Editor Report · Acceptance letter]

2 Feb 2022

PONE-D-21-18797R1 

Three-dimensional heads-up surgery in ab-interno trabeculotomy: Image processing-assisted trabeculotomy 

Dear Dr. Fujishiro:

I'm pleased to inform you that your manuscript has been deemed suitable for publication in PLOS ONE. Congratulations! Your manuscript is now with our production department. 

Kind regards, 

on behalf of

Dr. Aparna Rao 

Academic Editor

PLOS ONE